# Temporal Evolution of Seabed Scour Induced by Darrieus-Type Tidal Current Turbine

**Chong Sun [1], Wei Haur Lam [1,2,\*], Su Shiung Lam [3], Ming Dai [4] and Gerard Hamill [5]**

[1] State Key Laboratory of Hydraulic Engineering Simulation and Safety, Tianjin University, Tianjin 300350, China; chong@tju.edu.cn

[2] First R&D Services, A-08-16 M Suites, 283 Jalan Ampang, Kuala Lumpur 50450, Malaysia

[3] Pyrolysis Technology Research Group, Eastern Corridor Renewable Energy Group (ECRE), School of Ocean Engineering, Universiti Malaysia Terengganu, Kuala Nerus 21030, Malaysia; lam@umt.edu.my

[4] School of Engineering, University of Plymouth, Plymouth PL4 8AA, UK; Y.Dai@plymouth.ac.uk

[5] School of Natural and Built Environment, Architecture, Civil & Structural Engineering and Planning, Queen's University Belfast, David Keir Building, Stranmillis Road, Belfast BT9 5AG, UK; g.a.hamill@qub.ac.uk

\* Correspondence: joshuawhlam@hotmail.com or wlam@tju.edu.cn

**Abstract:** The temporal evolution of seabed scour was investigated to prevent damage around a monopile foundation for Darrieus-type tidal current turbine. Temporal scour depths and profiles at various turbine radius and tip clearances were studied by using the experimental measurements. Experiments were carried out in a purpose-built recirculating water flume associated with 3D printed turbines. The scour hole was developed rapidly in the initial process and grew gradually. The ultimate equilibrium of scour hole was reached after 180 min. The scour speed increased with the existence of a rotating turbine on top of the monopile. The findings suggested that monopile foundation and the rotating turbine are two significant considerations for the temporal evolution of scour. The scour depth is inversely correlated to the tip-bed clearance between the turbine and seabed. Empirical equations were proposed to predict the temporal scour depth around turbine. These equations were in good agreement with the experimental data.

**Keywords:** seabed scour; tidal current turbine; ocean renewable energy

## 1. Introduction

Tidal current energy is a proven type of ocean renewable energy gaining more attention after the successful installation of the first grid-connected tidal turbine in Strangford Loch, Belfast [1]. The vertical-axis tidal current turbine, commonly known as the Darrieus turbine, is able to capture the tidal current from various flow directions without changing turbine facing as horizontal-axis turbine [2]. Scour induced by turbines is an important engineering problem in tidal power projects. Dey et al. [3] reported the damages of scour around offshore structures. The contraction of flow by the turbine rotor changes the flow patterns around the turbine and accelerates the flow between the turbine and seabed due to blockage effects [4]. The dominant feature of scour around turbines is the horseshoe vortex system when monopile foundation is installed. The strength of the horseshoe vortex system is heavily affected by sediment bed velocity [5]. However, the slipstream below turbine can be accelerated due to the compression effect of turbine rotor. The development of scour hole induced by tidal current turbine amplifies the flow speed compared to the bridge pier scour due to flow suppression in the narrow area between rotor tip and seabed. This phenomenon has been proved in paper [6]. Hence the tidal turbine-induced scour is more complex.

Researchers simplified the tidal turbine-induced scour to propose the bridge pier scour equations without consideration of the influences of rotating turbine [7]. Seabed scour around a tidal current turbine can be investigated based on the known knowledge in bridge pier scours. Pier-induced scour focuses on the maximum scour depth, temporal scour evolution, flow field around scour hole, and parametric study on the scour process [5]. These scour works can be found in Sumer and Fredsøe's book "The Mechanics of Scour in the Marine Environment" [8] and Whitehouse's book "Scour at Marine Structures" [9]. The empirical equations for scour around bridge piers can be used for scour prediction around turbine [6]. However, the influences of the turbine on the evolution of scour hole are still unknown.

The horseshoe vortex system and downflow near front side of turbine foundation are the main factors for seabed scour. The horseshoe vortex system greatly strengthens the seabed shear stress inside the vortex's action area [10]. The initial scour hole occurs at both sides of the foundation where the shear stress of the seabed is greater than the critical shear stress. During the scour process, the sediment particles inside the hole are washed away downstream under the exerted action from horseshoe vortex. Rate of sediment scour decreases with the continuous reduction of vortex induced forces in time and finally reached the equilibrium state. On the other hand, the wake–vortex system can expand the scour hole downstream. Large scour holes may develop downstream by the combined action of horse-shoe vortex system and wake–vortex system. The wake–vortex system acts like a shovel to remove the bed material which is then carried downstream by shedding eddies from the supporting piles [11]. Bianchini et al. [12] analyzed the wake structure of vertical axis tidal current turbine using CFD (computational fluid dynamic) method and suggested the main vertical structures were generated by the rotors. Chen and Lam [4] found that tip clearance between the turbine and seabed is the important parameter to determine its impact to the scour by CFD simulation using OpenFOAM. They found the presence of turbine rotor changed the boundary layer profile and results in the altering of horseshoe vortex formation. The axial velocity of flow increased by ~10% of the initial velocity, which can cause deeper scour hole. Axial flow acceleration occurs between the turbine and seabed, but limited discussion was made to unveil the scour mechanism and these works are focused on the horizontal-axis turbine induced scour. Wake patterns of turbines are important foundations to investigate flow near seabed under turbine rotor. Wang et al. [13] measured the slipstream wash in terms of field velocities in turbine wake and surrounding flow, and suggested the turbine should be installed at least one diameter height to decrease its impact on seabed. Myers and Bahaj [14] demonstrated that increased velocity deficits and turbulence exist along the rotor centerline. Pinon et al. [15] used "vortex method" to simulate the flow field of horizontal-axis tidal current turbine. The "vortex method" was a velocity-vorticity numerical implementation of the Navier–Stokes equations to compute an unsteady evolution of the turbine wake by some three-dimensional software. The results show that flow velocity reduced behind rotor and gradually recovered downstream. Lam et al. [16] proposed a wake equation to illustrate the wake distribution of horizontal-axis turbine based on axial momentum theory and the experimental data. Flow acceleration can be clearly observed close to the rotating blade tips. Ma et al. [17] proposed a wake model for vertical-axis tidal current turbine. In their model, the minimum velocity position in lateral wake distribution deviated from centerline at efflux plane. The turbine's influence on the flow field reduced with the increase of the axial distance in turbine wake.

For the investigation of scour induced by tidal current turbine, Neill et al. [18] found the energy extraction from tidal current turbine reduced the overall bed level change; the asymmetry tidal region had a greater (20%) increase in sediment transport level compared to the tidal symmetry region. Hill et al. [19] carried out the experiments to examine the seabed scour around the foundation of horizontal axis tidal current turbine on an erodible channel. The works stated the scour development was accelerated compared to the scour around piles when rotor being installed upstream. Zhang [6] completed the numerical simulation to predict the horizontal-axis tidal current turbine scour evolution. Giles et al. [20] investigated the effect of foundation on the scour protection. Regarding hydrodynamic characteristics, Antheaume et al. [21] suggested that the efficiency of a single hydraulic Darrieus

turbine could be greatly increased in farm arrangements. The results proved that the hydrodynamic performance of Darrieus-type tidal current turbine could affect the flow field significantly. Recently, Sun et al. [22] proposed an empirical model to predict equilibrium scour depth around Darrieus tidal turbines in water. The temporal evolution of scour depth is important for engineering projects of tidal current turbine. Current works provide empirical equations to predict the temporal evolution of scour around foundation of Darrieus-type tidal current turbine.

## 2. Methodology

### 2.1. Experimental Set Up

Experiments were conducted in Marine Renewable Energy Laboratory, State Key Laboratory of Hydraulic Engineering Simulation and Safety, Tianjin University. The experiments used recirculating flume to simulate the tidal current flow and a 3D printing facility to create the tidal current turbine. Sandbox was used to simulate the seabed condition with the selected sand size to simulate the tidal turbine-induced seabed scour. A laser distance meter was used to measure the scour depth at designated positions of the scour regions. The overall experimental installations are shown in Figure 1. To make the experimental designs more clear, the overall diagram is shown in Figure 2.

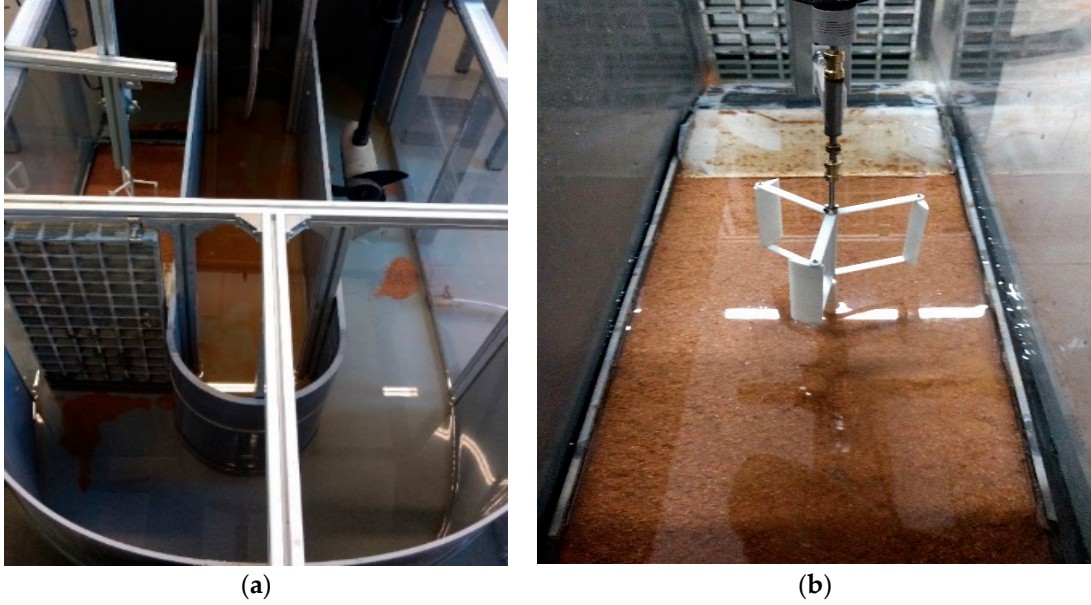

|     |     |
| :-: | :-: |
| (**a**) | (**b**) |

**Figure 1.** Experimental installation. (**a**) Overall setup of recirculating water flume [22] and (**b**) local setup of scour experimental part.

In the turbine scour experiment, the flume is 0.6 m in height and 0.35 m in width and the central cross-section is 0.8 m in width and 0.6 m in height. The turbine models were created by 3D printing. During the experiment process, flow circulates in a flume. Water is driven by propeller and moves towards the experimental side of flume. Flow moves through the flow-equalizing equipment and becomes more uniform. Then, the flow passes the tidal current turbine and erodes the sediment around supporting piles of turbine. The rotated turbine rotor is driven by Miniature DC rotor to ensure stable speed; the profiles of real-time seabed were measured by laser distance meter with 0.2 mm accuracy. During the experiment, the measurement grid is 5 mm in either the x-axis or y-axis. The seabed profile is measured in real time: the laser distance meter is installed above the water. It can launch laser and measure the distance between target and source by calculating the time difference. During the experiment, the flow moving is stopped by turning off the propeller after certain operational time. The overall sand bed profile is measured by Laser distance meter. Then reopening the propeller and

continuing scour experiment until the scour equilibrium. Laser-based instruments provide accurate results when they are utilized to acquire scour pattern. For instance, Dodaro et al. (2014) [23] and Dodaro et al. (2016) [24] employed a 3D laser scanner to measure scour profile downstream of a rigid bed.

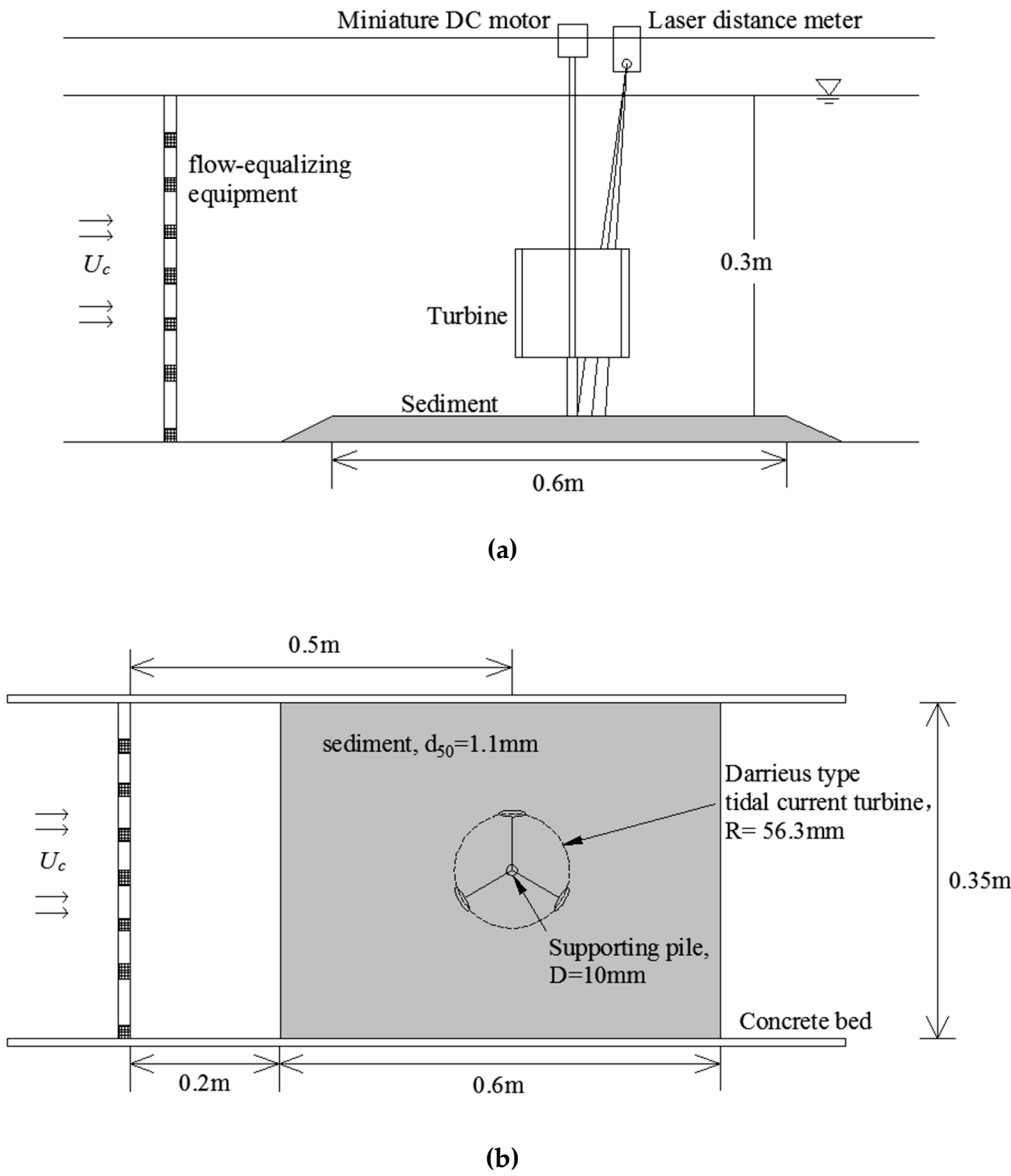

**Figure 2.** Overall design diagrams of scour experiment: (**a**) front view and (**b**) overhead view.

Various parameters were considered in the series of experiments to investigate the influence factor of turbine scour. The constant parameters are supporting pile diameter, rotor height, blade chord length, water depth, inlet velocity, sediment diameter, and rotational speed. The variable parameters are rotor radius and tip-bed clearance. These two parameters are main variables in our research. The specific value of each parameter is shown in Table 1. It is worth noting that this recirculating flume was used in the previous works to propose empirical model for equilibrium scour depth induced by turbine [22].

**Table 1.** Experimental parameters.

| Parameters | Values |
| --- | --- |
| Rotor radius $R$ (mm) | 56.3, 45.9, 37.4 |
| Rotor height $H$ (mm) | 87.5 |
| Chord length $c$ (mm) | 20.4, |
| Dimensionless tip clearance $C/H$ | 0.25, 0.5, 0.75, 1 |
| Monopile diameter $D$ (mm) | 10 |
| Rotational speed $w$ (rpm) | 110 |
| Depth of water $h$ (m) | 0.3 |
| Width of flume $b$ (m) | 0.35 |
| Sediment diameter $d_{50}$ (mm) | 1.1 |
| Mean current velocity $U_c$ ($m/s$) | 0.23 |

*2.2. Scaling Effects*

2.2.1. Physical Similarity Relationships

The size of the turbine model is constrained by the dimensions of the recirculating flume. A turbine height of 87.5 mm is chosen so it could occupy ~1/3 of 0.3 m water depth. The size of the turbine is 1:8 of the physical model in paper [25]. Rotor radii of 56.3, 45.9, and 37.4 mm were set-up to ensure the solidity of turbine within the scope of 1.09 to 1.64 to investigate the impact of rotor radius on turbine scour. This solidity has been proved to show great energy extraction efficiency [25]. The turbine model represents a ratio of approximately 1:60 of full scale Darrieus-type tidal current turbine like Kobold project [26]. The ratio of turbine height and water depth was the same as Kobold. The inlet velocity is set as 0.23 m/s ensuring the clear water scours at initial scour stage. In addition, the tip speed ratio (TSR) is an important dimensionless number to influence the hydrodynamic performance of Darrieus-type tidal current turbine. The TSR is designed to be same as paper [25] to make sure the turbine has enough energy extraction efficiency. The TSR can be calculated by Equation (1):

$$\text{TSR} = \frac{\omega R}{U_c} \tag{1}$$

The Reynolds number of the turbine wake induced by the Darrieus-type turbine model used in the experiment was ~287,500, which is calculated using Equation (2).

$$\text{Re} = \frac{\rho U_c D_t}{\mu} \tag{2}$$

where $\rho$ is density of water (kg/m$^3$), $U_c$ is mean current velocity (m/s), $D_t$ is turbine diameter (m), and $\mu$ is the dynamic viscosity of water (*Pa s*). Rajaratnam [27] suggested the effect of viscosity could be neglected as long as the Reynolds number of propeller jet was more than 10,000. Therefore the viscosity effect could be neglected in the analysis of our turbine model.

2.2.2. Scaling Effects of Experimental Setup

The mean sediment diameter was kept at 1.1 mm by filtering the layers of the sieves in the current experiment. The scour depth can be impacted by the critical startup shear stress of sediment with different diameter [10]. However, our research focused on the influence of turbine's tip clearance and rotor radius on temporal scour depth; the impact of sediment diameter is not a main variable. In previous experimental study of scour, the sediment size $d_{50}$ was the same size as naturally occurring sediments—from 0.5 mm to 2.0 mm—such as [18,28]. We choose 1.1 mm sediment size as a normal prevailing condition to ensure that the sand is under clear water scour at the initial stage, in line with reality.

The size of water flume used in our experiment is shown in Figure 2. The fixed wall surfaces on both sides of flume have influence the flow distribution and experimental accuracy. The flume width is

0.35 m, which is ~6 times than turbine radius and 35 times the supporting pile diameter. In Roulund's experiment of scour around pile [29], the flume width was 36 times than pile diameter, which is almost same as our flume. In addition, based on existed experimental research of flow field around tidal current turbine [16], the operational turbine can disturb flow around it, but shows little impact on flow more than 3R distance in radial direction. Hence the flume is wide enough for our experiments.

During the experiments, the incoming velocity is maintained as 0.23 m/s measured by pitot tube in experimental flow region. This flow velocity is less than critical current speed. The formula to calculate the critical current speed can be found in paper [11]. It appears as clear water scour at first, but the initial flow cannot sweep the sediment and the scour phenomenon only occurs around the turbine's supporting pile. It should be pointed out that due to our innovative type of horizontal recirculating flume the flow on the outside of flume is a little quicker than the inside flow at the corner. When the flow moves out of the corner and towards the experimental area, the flow-equalizing equipment can reduce the speed difference and turbulence intensity to make the flow more stable. In addition, the distance between the turbine center and flow-equalizing equipment is ~10*R*. This distance is long enough to minimis the influence of uneven velocity distribution and produce uniform flow in the measurement area. To verify the reliability of flow uniformity in measurement area, the 3D shape of scour hole and the sand dune downstream is shown in Figure 3. In Figure 3, the flow moves from right to left. The white pile is supporting pile of turbine. The outline of scour hole and sand dune is indicated by white line and the centerline of sand dune is sketched by black line. The scour shape shows great symmetry downstream. However, the outside tail of the sand dune is a little longer than the inside tail. This is due to the wake asymmetry of Darrieus-type tidal current turbine which has been proposed in paper [17].

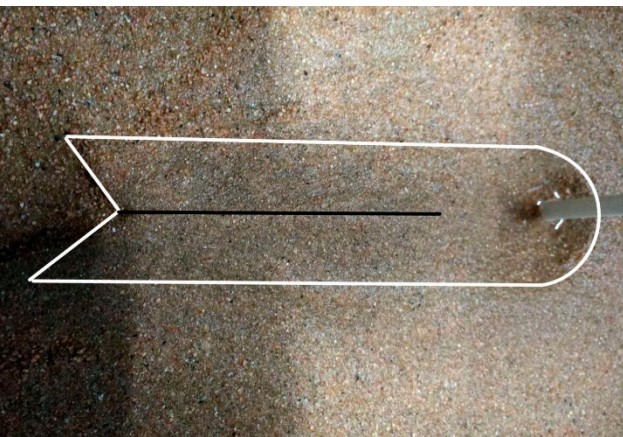

**Figure 3.** The sediment bed after scour equilibrium in case *C/H* = 1.0, *R/D* = 5.63.

In summary, the turbine model used in our experiments is small scale model compared with real tidal current turbine under the constraints of the experimental setup. However, the hydrodynamic performance of the turbine can be guaranteed based on reasonable TSR and turbine solidity. Meanwhile, the recirculating flume is wide and long enough to ignore the wall boundary's impact on scour process, and ensures the accuracy of experiments. Our primary purpose is to investigate the temporal development of scour depth induced by turbine with different rotor radius and tip clearance, and our experimental conditions are qualified for this work.

## 3. Temporal Variations of Scour Profiles

Darrieus turbine induced scour has limited data compared to the horizontal turbine induced scour. Previous works simplified the rotating turbine in the scour predictions [20]. Tip-bed clearance and the turbine radius are two main parameters of turbine to influence scour processes. In the current works, the temporal scour evolution are discussed in Section 3 with particular focusses on the influences of

tip-bed clearance in Section 3.1 and turbine radius in Section 3.2. The turbine-induced contracted flow leading to scour is considered to occur on top of the monopile scour due to horseshoe effects.

Selected cases with various tip-bed clearances and turbine radii were tested to produce data to investigate the impact of Darrieus turbine to seabed scour. Seabed scour reached equilibrium after ~150 min in the experiments. The temporal variations of scour profiles are presented at 30 min, 90 min, and 150 min. Tip-bed clearance (*C*) was nondimensionalized by using the turbine height (*H*). In the first four cases, the turbine rotor remained 56.25 mm. The equilibrium scour hole depth is 2.2*D*, 2.2*D*, 1.7*D*, and 1.6*D* in cases with different tip clearance (*C/H* = 0.25, 0.5, 0.75, and 1.0 respectively). In other two cases, the tip clearance is remained as 0.5H, and the equilibrium scour hole depth reaches 2*D* when *R* = 45.9 mm, and 2.6*D* when *R*= 37.4 mm.

### 3.1. Temporal Seabed Scours at Various Tip Clearance

The measured temporal developments of the scour profiles of turbine are presented in Figures 4 and 5 at different tip clearances. Four tip-bed clearances of *C/H* = 1, 0.75, 0.5 and 0.25 respectively were examined. The nondimensional temporal variations of the scour profiles at 30 min, 90 min and 150 min are plotted in the figures. Figure 4 shows scour profiles at centerline of supporting pile (y = 0), and Figure 5 shows scour profiles at y = 0.5*D*. For the x-axis, the x/*D* = 0 is located at monopile central line. For y-axis, *S/D* is the scour depth nondimensionalized by monopile diameter (*D*) in the vertical direction.

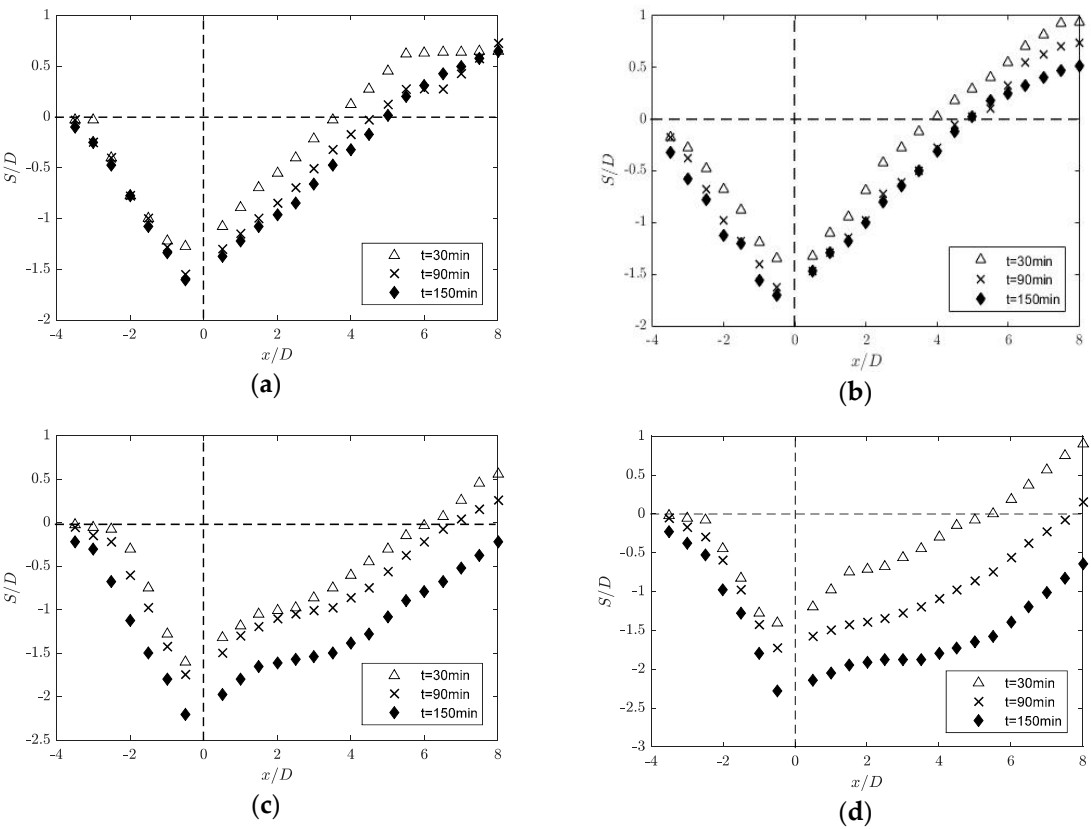

**Figure 4.** Temporal seabed profiles with different tip clearance at location y/D = 0. (**a**) Tip clearance =1H, (**b**) tip clearance = 0.75H, (**c**) tip clearance = 0.5H, and (**d**) tip clearance = 0.25H.

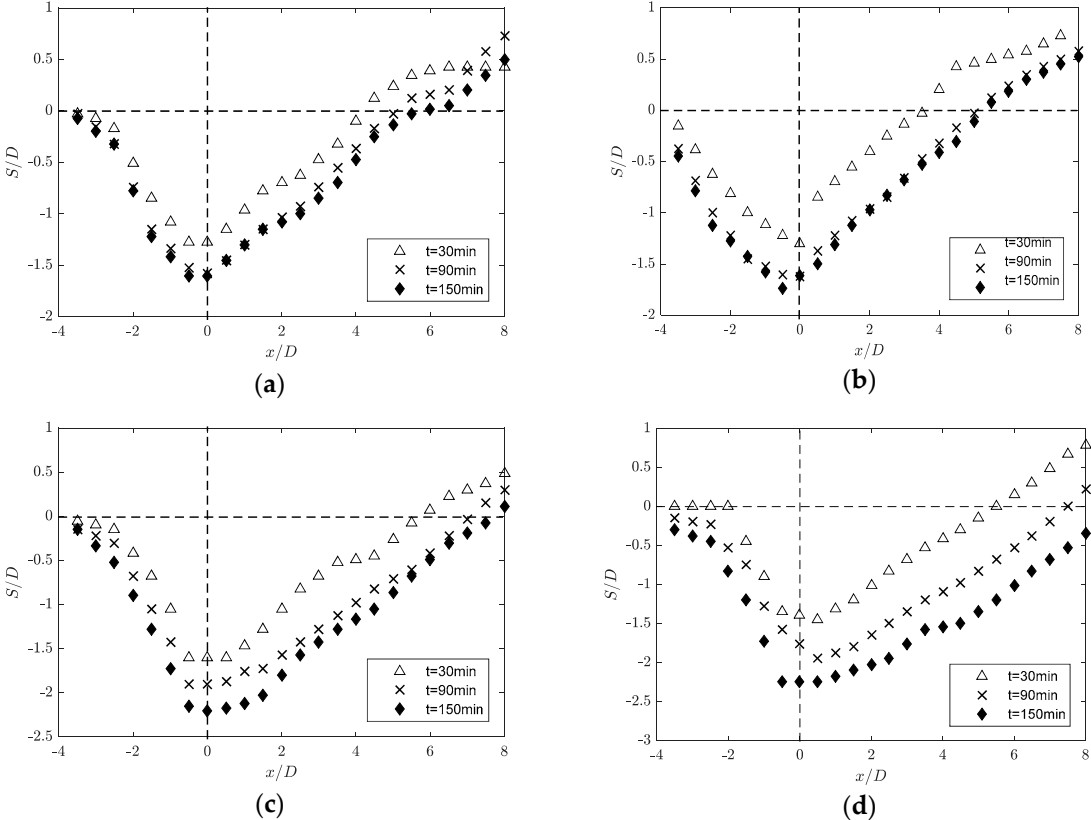

**Figure 5.** Temporal seabed profiles with different tip clearance at location y/*D* = 0.5. (**a**) Tip clearance = 1*H*, (**b**) tip clearance = 0.75*H*, (**c**) tip clearance = 0.5*H*, and (**d**) tip clearance = 0.25*H*.

### 3.1.1. Size of Scour Hole

The size of the scour holes increases with time up to 150 min with limited impact of tip-bed clearances. Tip-bed clearance gives significant impacts to the scour depth (*S*/*D*), but insignificantly influences the growing trend of the size of scour hole. The size of scour hole increases over time from 30, 90, and 150 min at all tip clearance *C*/*H* = 1, 0.75, 0.5, and 0.25. In all cases, the scour holes were rapidly developed in the beginning process. Temporal scour depth can reach 75% of the final depth after initial 30 min in the model tests. The increase rate of scour depth slows down with time. After scour equilibrium, the scour depth no longer increases. However, the horizontal size gradually increases with time. For the clearances at *C*/*H* = 0.25 and 0.5, the deposition moves back quickly, and the deposition mound cannot be observed in the range of −4*D*–8*D* along x-axis after 150 min. The deposition wedge moves backwards during the scour process.

### 3.1.2. Maximum Scour Depth

Scour depth increases with time for Darrieus tidal turbine at various tip-bed clearances. After equilibrium, the maximum scour depths are 2.2*D*, 2.2*D*, 1.7*D*, and 1.6*D* for *C*/*H* = 0.25, 0.5, 0.75, and 1.0, respectively. It can be shown that maximum scour depths in cases of *C*/*H* = 1.0 and *C*/*H* = 0.75 are much smaller than cases of *C*/*H* = 0.5 and *C*/*H* = 0.25. The difference is ~23%. The reason for this is when the turbine is installed higher than 0.5H distance from bed, the scour process is clear water scour. The rotating rotor shows weak influence on scour around the turbine's foundation. The slipstream velocity between seabed and turbine is not great enough to move sediment around scour hole. The scour hole developed mainly due to the action of a horseshoe vortex system. However, when the turbine is installed near the bed (for cases of *C*/*H* = 0.5 or 0.25 in the experiment), the rotor's effect is notable. The near bed velocity is great enough to move sediment around hole. The scour process is combined with digging by the downflow and slant bed erosion until the dynamic equilibrium is reached.

### 3.1.3. Position of Maximum Scour Depth

The position of maximum scour occurs at both back sides of the monopile regardless of the clearance distances. The specific location is x/D = −0.5, y/D = 0.5 in each case. This position is also the maximum bed shear stress occurs under the action of horseshoe vortex system. These results also prove that the horseshoe vortex system still the main function for scour around tidal current turbine.

### 3.1.4. Deposition

The deposition of sediment can be seen in Figure 6: a dune formed behind the foundation. The dune moved slowly downstream with time. When C/H = 0.75, the axial scour hole edge along the centerline at the location of approximately x = 4.5D at 90 min. However the hole edge migrated to a location approximately x = 7.8D downstream after 90min when C/H = 0.25. The migrating speed of dune is inverse proportional with the clearance between seabed and rotor. In each case, the highest sand dune is ~1D. The axial location of the highest sand dune migrates with time. The upstream scour hole slope is much steeper than the downstream slope.

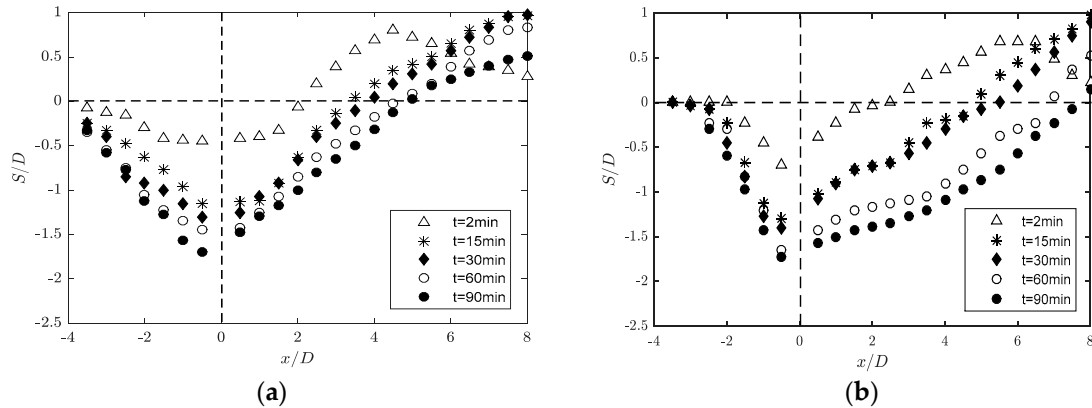

**Figure 6.** Temporal seabed profiles of different times (t = 2 min, 15 min, 30 min, 60 min, and 90 min) with different tip clearance at location y/D = 0. (**a**) Tip clearance =0.75*H* and (**b**) tip clearance = 0.25*H*.

### 3.1.5. Remark

In general, the maximum scour hole depth occurs at the back side of the foundation. The location is approximately x/D = −0.5, y/D = 0.5. A scour hole is developed around the supporting pile and the deposition mound occurs behind the turbine. The depositional wedge moves further downstream. The hole size shows an inverse correlation with the space between turbine and seabed. The deposited mount moves faster with lower installation height. Furthermore, when the tip clearance is high enough, the turbine has little impact on seabed sour development. The evolution speed of scour shows great increase with a lower installation height. And this influence increases until the appearance of live bed scour. The scour development shows little change with the continuous decrease of tip clearance in live bed scour process. This process shows the same trend with live bed scour at piers [30].

### 3.2. Temporal Seabed Scours at Various Rotor Radius

The influence of rotor radius is investigated at radii of 37.4 mm, 45.9 mm, and 56.3 mm, as shown in Figures 7 and 8. Figure 7 shows temporal scour profiles at y = 0, and Figure 8 shows temporal scour profiles at y = 0.5*D*.

As discussed in the previous section, tip-bed clearance gives significant temporal impacts to the size of the scour hole, maximum scour depth, and position of the maximum scour. The constant tip-bed clearance C/H = 0.5 was chosen with the changes of turbine radius. A tip-bed clearance of C/H = 0.5 is commonly used distance in the installation.

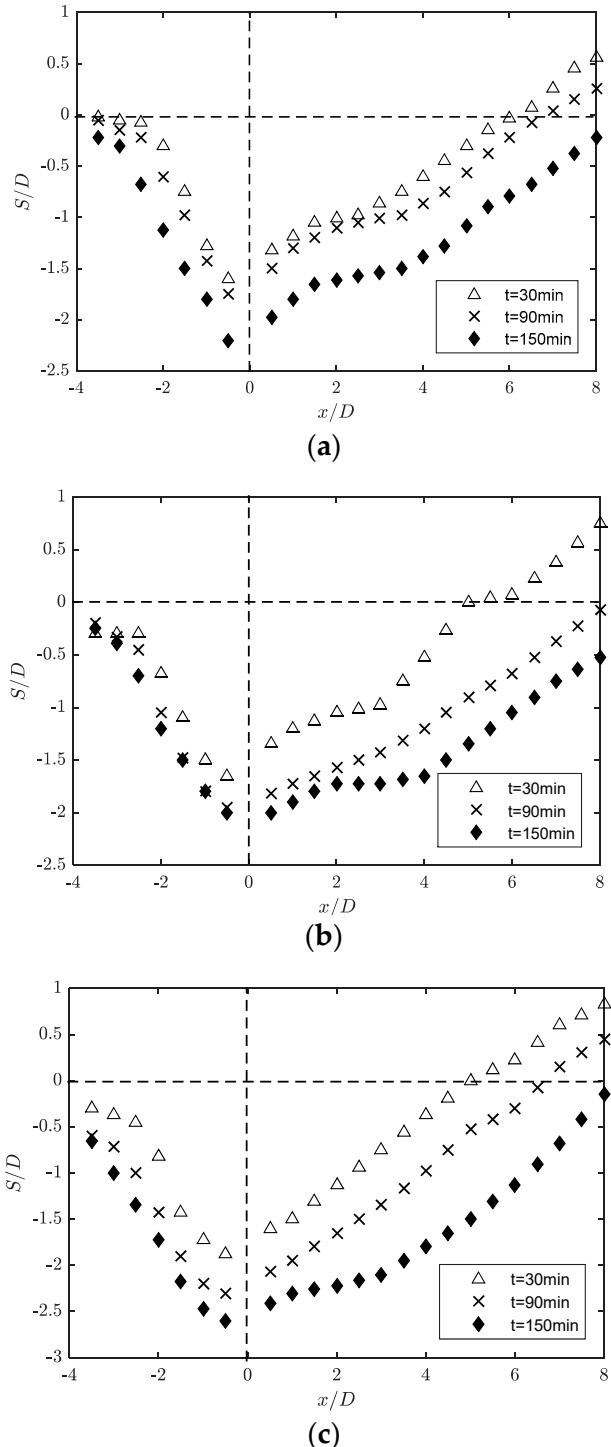

**Figure 7.** Temporal seabed profiles at different times with different turbine radius at location y/0. (**a**) *R* = 56.3 mm, (**b**) *R* = 45.9 mm, and (**c**) *R* = 37.4 mm.

Rotor radius is an important factor for hydrodynamic performance and wake structure of Darrieus-type tidal current turbine [26]. The energy extraction efficiency is closely related to the tip speed ratio which is a function define by rotor radius, inlet velocity, and rotational speed. The scour process around the turbine foundation is greatly influenced by the turbine energy extraction [18]. Therefore, the rotor radius has significant impact on the temporal evolution of scour depth induced by tidal current turbine.

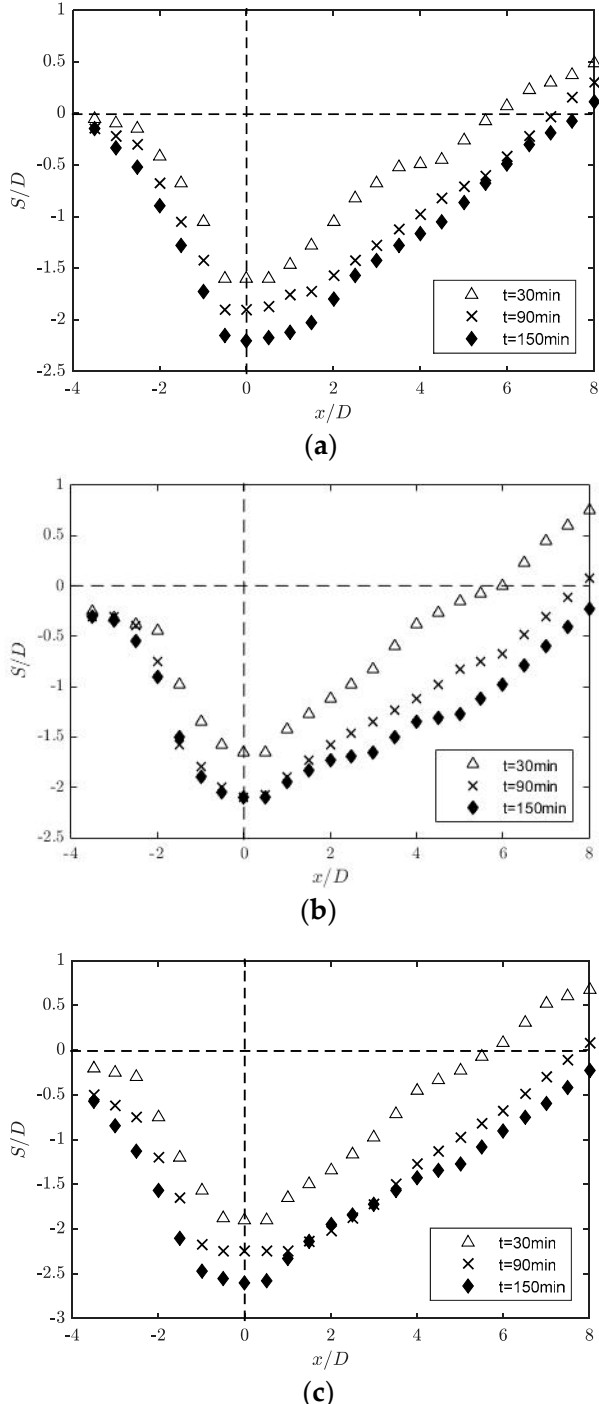

**Figure 8.** Temporal seabed profiles at different times with different turbine radius at location y/D = 0.5. (**a**) *R* = 56.3 mm, (**b**) *R* = 45.9 mm, and (**c**) *R* = 37.4 mm.

### 3.2.1. Size of Scour Hole

The scour process is live bed scour in all the tested cases with different rotor radius. The size of scour hole increases with the time until equilibrium state at 150 min. The temporal seabed profiles at 30 min, 90 min, and 150 min can be seen in Figures 7 and 8. The size of scour hole shows a nonlinear relationship with rotor radius. For instance, the maximum size of the scour hole is obtained at *R* = 37.4 mm, and the minimum size of scour hole is at *R* = 45.9 mm. In the case of *R* = 56.3 mm, a medium-sized equilibrium scour hole is obtained.

### 3.2.2. Maximum Scour Depth

After scour equilibrium, the maximum scour depths were 2.7*D*, 2*D*, and 2.2*D* for *R* = 37.4 mm, *R* = 45.9 mm, and *R* = 56.3 mm, respectively. The maximum scour depth decrease first, and then increases with the increase in rotor radius. The results can be explained that the change of energy extract efficiency of turbine is nonlinear against tip speed ratio. In the experiment, the rotational speed and inlet velocity is remained constant. Hence, the tip speed ratio increases linearly with rotor radius. Accordingly, the equilibrium scour depth shows a complex nonlinear relationship with rotor radius.

### 3.2.3. Position of Maximum Scour Depth

According to the model tests, the position of the maximum scour occurs at both back sides of the monopile regardless of the rotor radius. This result agrees with previous researches of scour at piers [28] and scour induced by turbine [18]. The specific location is x/*D* = −0.5, y/*D* = 0.5 in each case. These results are same as the cases with different tip clearance. The maximum scour depth position is still decided by horseshoe vortex system. In each case, the equilibrium deposition of sediment cannot be observed within 8D downstream. However the moving process of sand dunes behind the turbine's foundation can be seen at different scour time.

### 3.3. Overall Temporal Evolution of Scour Depth

The temporal evolution of scour depths at different rotor radius and tip clearance are plotted in Figure 9. In each case, the scour hole develops rapidly in the beginning process and then grows gradually until 150 min. At first, at approximately 10 min, the scour depth increases rapidly. The general appearance of scour around piles: turbine scour starts with an increase of flow speed around turbine supporting pile. The sediment is easily washed away by large bed shear stress on both sides of pile. The scour hole expands by the energy of downflow and horseshoe vortex. However, the less bed shear stress and weaker energy of horseshoe vortex exists inside the hole along with the development of scour hole until the equilibrium situation. The maximum final scour depth shows two scour types against different tip clearance when the turbine radius remains unchanged. When *C/H* > 0.5, the equilibrium scour hole depth *S/D* is less than 1.7. An equilibrium scour depth is reached after ~150 min of run time. However, when *C* ≤ 0.5, the equilibrium scour depth is ~2.2*D*, and the scour equilibrium is reached after ~110 min. The reason is illustrated below.

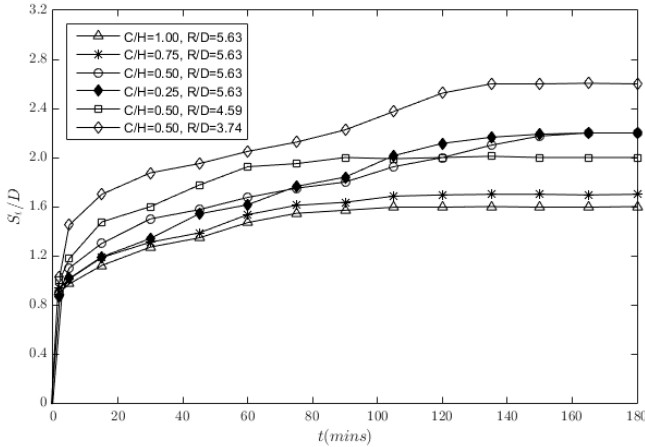

**Figure 9.** Evolution of maximum scour depth in time with different rotor radius and tip clearance.

For all model tests, even though each test has the same inlet velocity, the bed velocity is different due to different setup of turbine model. The scour depth around pile increases with the increase of flow velocity near sediment bed in clear water scour, but the depth shows little increase when the inlet velocity is faster than critical velocity [11]. In cases 1–4 with same rotor radius but different tip

clearance, flow moves faster at lower tip clearance generally. This is due to greater flow acceleration below the turbine at lower tip clearance, thus the velocity of downflow and near-bed flow is amplified. Hence the evolution of the scour hole can be sped up. In cases 3, 5, and 6 with the same tip clearance but different rotor radius, there is also a big difference in scour depth. When dimensionless tip clearance is maintained as *C/H* = 0.5, the equilibrium scour hole depth reaches a maximum value of 2.6*D* when *R/D* = 3.74, and minimum value of 2.0*D* when *R/D* = 4.59. As discussed in Section 3.2, this result reflects the impact of the turbine's hydrodynamic performance on the seabed scour.

To further illustrate the shape of scour hole impacted by turbine's set up, the temporal development of scour width is shown in Figure 10. Final scour hole width is located between 3.5*D* to 4.8*D* with different rotor radius and tip clearance. It can be seen that final hole width shows same trends as scour depth, the max hole width occurs in case *C/H* = 0.50, *R/D* = 3.74. In each case, the final scour hole width is ~2 times that of the scour depth. This can be explained by the development mechanism of the scour hole. The horizontal expansion of the scour hole is a result of a sand slide. With the sustained evolution of scour depth, the slope of hole exceeds sediment repose angle, hence the sand at edge of hole slide into the scour hole. Hence the equilibrium scour hole slope is fixed, which is approximately the sediment repose angle. Thus the horizontal and vertical dimensions of the scour hole show a 2:1 ratio.

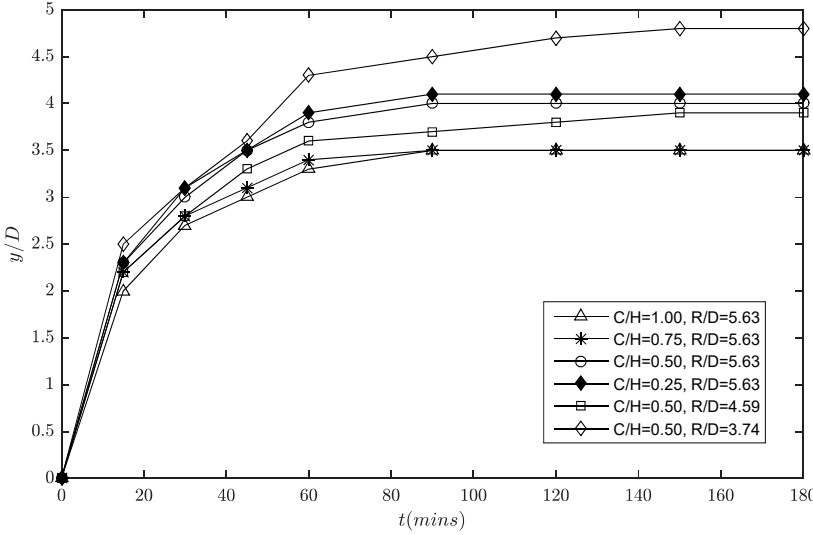

**Figure 10.** Evolution of scour hole width in time with different rotor radius and tip clearance.

## 4. Equations to Predict Temporal Evolution of Scour Depth

Tip-bed clearance and turbine radius are used to propose novel empirical equations used to predict Darrieus turbine-induced scour. Temporal scour equations are compared to the established works from bridge pier scour equations, ship propeller scour equations, and relevant tidal turbine scour data.

### 4.1. Derivation of the Temporal Scour Equations

Previous researchers proposed the importance of the scour and predict the temporal variation of scour depth [31–36]. The temporal scour depth $S_d$ was predicted by a selected percentage of equilibrium scour depth $S_e$ with consideration of the effects of current velocity, pier diameter, and water height. The selected scour equations to compare the proposed temporal scour depth model are listed below. Several numerical equations were found to predict the scour depth induced by propeller jet [37,38]. The flow velocity, propeller diameter, tip clearance, and sediment diameter are the main factors that influence the evolution of the scour hole. During the scour process induced by turbine, the temporal scour hole is deeper and wider compared to the pier-induced scour due to the flow contraction caused

by rotating turbine, the increase ratio is ~10–75% with various turbine rotor radius and tip clearance in the model tests. However the scour depth is smaller than the propeller jet scour. Therefore, the pier scour equations underpredict the turbine-induced scour, whereas the propeller scour equations overpredict the temporal scour depth induced by the turbine. Engineers do not have the appropriate empirical equations to predict the temporal development of scour hole around turbine foundation. Accordingly, this paper proposed more applicable empirical equations to predict temporal evolution of scour depth induced by tidal current turbine.

For empirical equations to predict temporal scour depth at piers, Sumer et al. [31] proposed the Equation (3), where $S_e$ is maximum scour depth at equilibrium stage, $t_e$ is the equilibrium time of scour:

$$\frac{S_d}{S_e} = 1 - e^{\left(-\frac{t}{t_e}\right)} \tag{3}$$

Melville and Chiew [32] proposed the Equation (4), where $U_{cr}$ is critical velocity of sediment, $U$ is approach velocity of flow:

$$\frac{S_d}{S_e} = e^{\left\{-0.03\left|\frac{U_{cr}}{U_c}\ln\left(\frac{t}{t_e}\right)\right|^{1.6}\right\}} \tag{4}$$

Oliveto and Hager [33] proposed the Equation (5), where $T$ is dimensionless time of scour:

$$\frac{S_d}{L_R} = 0.068\sigma^{-1/2}F_d^{1.5}\log T \tag{5}$$

In Equation (5),

$$L_R = D^{2/3}h^{1/3}, \; F_d = \frac{U_c}{(g'd_{50})^{1/2}}, \; \sigma = (d_{84}/d_{16})^{0.5}, \; T = \left(\frac{(g'd_{50})^{0.5}}{L_R}\right)t, \; g' = \left[\frac{\rho_s - \rho}{\rho}\right]g.$$

Lança et al. [34] proposed the Equation (6), where $D$ is the diameter of pile:

$$\frac{S_d}{S_e} = 1 - e^{\left[-a_1\left(\frac{U_c t}{D}\right)^{a_2}\right]} \tag{6}$$

In Equation (6),

$$a_1 = 1.22\left(\frac{D}{d_{50}}\right)^{-0.764}, \; a_2 = 0.99\left(\frac{D}{d_{50}}\right)^{0.244}$$

Choi and Choi [35] proposed the Equation (7), where $h$ is flow depth:

$$\frac{S_d}{S_e} = e^{\left\{0.065\left(\frac{U_c}{U_{cr}}\right)^{0.35}\left(\frac{h}{D}\right)^{0.19}\ln\left(\frac{t}{t_e}\right)\right\}} \tag{7}$$

Harris et al. [36] proposed Equations (8) and (9), in these models, Breuser's [34] equation has been used to predict the maximum scour depth:

$$\frac{S_d}{S_e} = 1 - e^{\left(-\frac{t}{t_e}\right)^n} \tag{8}$$

$$\frac{S_e}{D} = 1.5K_sK_\theta K_bK_d\tanh\left(\frac{h}{D}\right) \tag{9}$$

In our equations, the parameters that influence the temporal scour depth $S_t$ are related in Equation (10) and when dimensionless lead to Equation (11).

$$S_t = f_1(U_c, D, C, d_{50}, R, g, \rho, \rho_s, h, t) \tag{10}$$

$$S_t = f_2\left(\frac{U_c t}{D}, \frac{C}{H}, \frac{d_{50}}{D}, \frac{R}{D}, \frac{U_c}{(gh)^{0.5}}, \frac{\rho_s - \rho}{\rho}\right) \tag{11}$$

Dimensionless tip clearance and turbine radius are important for scour process. $\rho_s$ is the sediment density, which is considered as constant in nature. By use of Vaschy–Buckingham theorem, Equation (12) is obtained.

$$\frac{S_t}{D} = f_3\left(\frac{U_c t}{D}, F_r, \frac{d_{50}}{D}, \frac{C}{H}, \frac{R}{D}\right) \tag{12}$$

$$F_r = U_c/(gh)^{0.5} \tag{13}$$

The Froude number ($F_r$) is constant when considering constant flow depth and uniform flow velocity, as calculated by Equation (13). $F_r = 0.13$ in the current works. The sediment diameter is remained as 1.1mm at seabed. Equation (12) can be simplified as Equation (14):

$$\frac{S_t}{D} = f_4\left(\frac{U_c t}{D}, \frac{C}{H}, \frac{R}{D}\right) \tag{14}$$

Live bed scour and clear water scour can be found in the experiment against different tip clearance. Clear water scour occurs when $C/H > 0.5$; the temporal scour depth is closely related to current velocity, foundation diameter, rotor radius, and tip clearance. When $C/H \leq 0.5$, live bed scour takes place around monopile foundation, and temporal scour depth has no relevance to tip clearance or rotor radius. Dimensionless analysis produces the empirical equation to predict the temporal evolution of scour hole depth around monopile foundation of Darrieus tidal current turbine in Equations (15)–(19):

a. For clear water scour ($C/H > 0.5$):

$$\frac{S_t}{D} = k_1\left[\log_{10}\left(\frac{U_c t}{D}\right) - k_2\right]^{k_3} \tag{15}$$

where

$$k_1 = 0.269\left(\frac{C}{H}\right)^{1.1}\left|\frac{R}{D} - 5\right|^{-0.09} \tag{16}$$

$$k_2 = -1.088\left(\frac{C}{H}\right)^{0.171}\left|\frac{R}{D} - 5\right|^{-0.3} \tag{17}$$

$$k_3 = 1.233\left(\frac{C}{H}\right)^{-0.7}\left|\frac{R}{D} - 5\right|^{0.15} \tag{18}$$

b. For live bed scour ($C/H \leq 0.5$):

$$\frac{S_t}{D} = 0.131\left[\log_{10}\left(\frac{U_c t}{D}\right) + 1.11\right]^{1.869} \tag{19}$$

Figure 11 shows the comparison between experimental data and data calculated by proposed temporal scour depth equation; $R^2$ is 0.93. The proposed model well agrees with the experimental data.

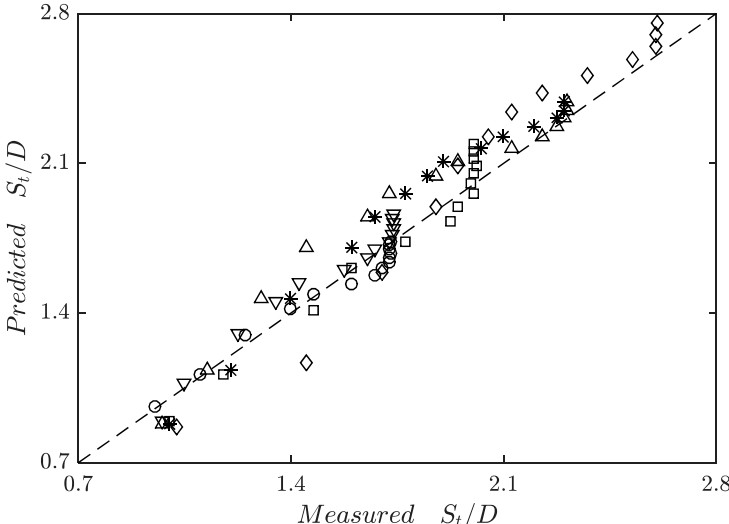

**Figure 11.** Comparison between observed and computed time-dependent scour depth.

*4.2. Comparison of the Proposed Model with the Previous Works*

The proposed empirical model was compared to the bridge pier scour model, propeller scour model, and the scour data of tidal turbine.

4.2.1. Comparison with Bridge Pier Scour Equations

Figure 12 shows the comparison between temporal evolution of scour depth around turbine foundation calculated by our proposed model and scour depth at bridge piers predicted by existing equations. In present study, six previously developed time-dependent scour depth equations [31–36] are chosen for checking the accuracy. According to Gaudio et al. (2010) [39] and Gaudio et al. (2013) [40], due to different mathematical structures in scour prediction formulas, the available bridge pier scour equations may give significantly different predictions.

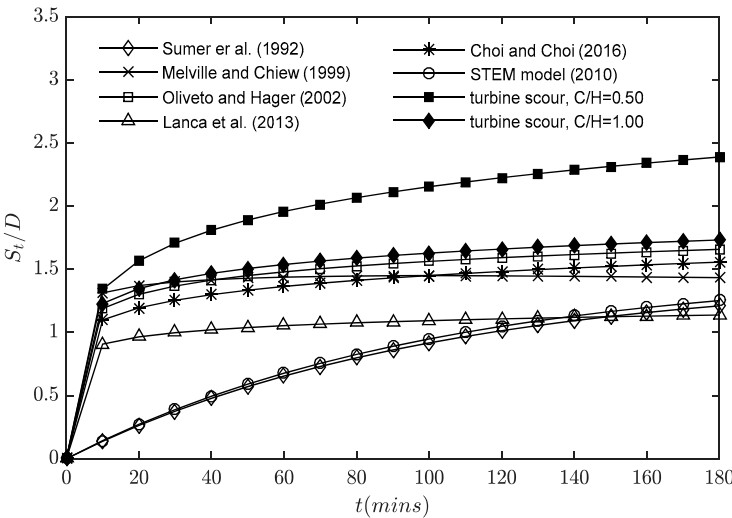

**Figure 12.** Comparison of time-dependent scour depth around turbine and scour at piers.

For comparison, the equilibrium scour hole depth $S_e$ in Equations (3)–(9) is set to be 1.45$D$. This is the equilibrium scour depth around single pile for the same test conditions measured in the current experiment. Two cases (*C/H* = 0.5 or 1) in the current experiment have been chosen to be references. These two cases are typical turbine scour process as live bed scour or clear water scour. Equations

(4), (5), and (7) show great agreement with the temporal scour depth at *C/H* = 1.0. It shows a great increase in scour depth in the first 10 min and low growth in the later scouring time. When *C/H* = 1.0, the seabed scour around turbine is more like scour at piers. The scour process is not much affected by turbine rotor. When the tip clearance lower than 0.5*H*, the evolution of scour hole is greatly affected by turbine rotor. The equilibrium scour hole depth is much more than other cases in Figure 12. The increasing extent is ~50%.

### 4.2.2. Comparison with the Equations of Ship Propeller Induced Scour

The temporal scour depth induced by turbine and propeller jet has been compared in Figure 13. Two empirical equations for temporal propeller jet induced scour depth prediction proposed by Qurrain [37] and Hong [38] are chosen to compare the scour induced by turbine. These equations are listed below. Temporal scour depth induced by turbine and propeller jet has approximate similar variation trend, as shown in Figure 13. The scour depth increases greatly in the first 10 min. The scour depth increases slowly until equilibrium. The scour mechanisms of these two types of scour are different. In the ship propeller jet-induced scour, the propeller jet causes high shear force which can activate sediment. The maximum scour depth occurs at the centerline of the propeller in a streamwise direction downstream. However, the dominant feature of scour around foundation of turbine is horseshoe vortex. The evidence of turbine rotor disturbs the surrounding flow and accelerates flow under the turbine. Hence the scour hole can be deeper than the scour hole around piers. The position of maximum scour depth is around supporting pile of turbine. Generally, the scour depth induced by propeller jet is much deeper than scour around tidal current turbine at same installation height. In Figure 13, the equilibrium scour hole depth predicted by Hong's empirical equation reaches about 3.5D when C/H = 0.5, which is 58% deeper than equilibrium depth around turbine.

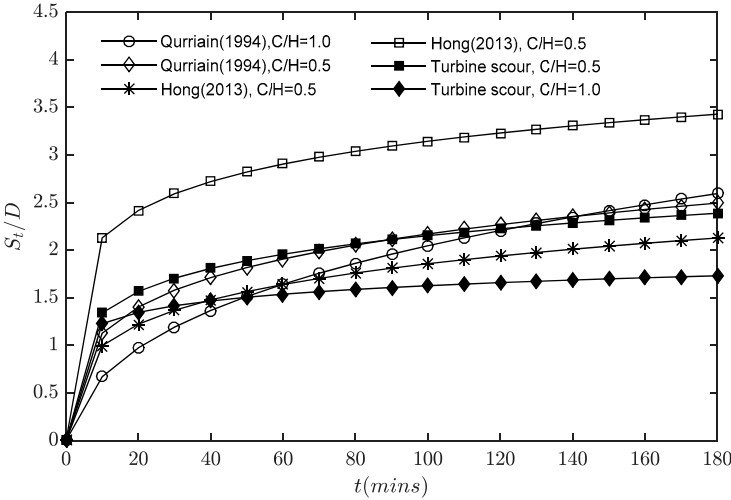

**Figure 13.** Comparison of time-dependent scour depth around the turbine with scour induced by propeller jet.

For empirical equations to predict temporal scour depth induced by propellers, Qurrain et al. [37] proposed Equation (20), where $S_p$ is maximum scour depth in mm at any time $t$, in seconds.

$$S_p = \Omega[\ln(t)]^\Gamma \tag{20}$$

In Equation (20),

$$\Omega = \left[\frac{C}{d_{50}}\right]^{-4.758}\left[\frac{D_p}{d_{50}}\right]^{2.657}[F_0]^{3.517}, \Gamma = \left[\frac{C}{d_{50}}\right]^{0.758}\left[\frac{D_p}{d_{50}}\right]^{-0.339}[F_0]^{-0.479}.$$

Hong et al. [38] proposed Equation (21), where $D_p$ is propeller diameter, $U_0$ is efflux velocity.

$$\frac{S_p}{D_p} = k_1 \left[ \log_{10} \left( \frac{U_0 t}{D_p} \right) - k_2 \right]^{k_3} \tag{21}$$

In Equation (21),

$$k_a = 0.014 F_0^{1.120} \left( \frac{C}{D_p} \right)^{-1.740} \left( \frac{C}{d_{50}} \right)^{-0.170}$$

$$k_b = 1.882 F_0^{-0.009} \left( \frac{C}{D_p} \right)^{2.302} \left( \frac{C}{d_{50}} \right)^{-0.441}$$

$$k_c = 2.477 F_0^{-0.073} \left( \frac{C}{D_p} \right)^{0.53} \left( \frac{C}{d_{50}} \right)^{-0.045}$$

### 4.2.3. Comparison with Published Data of Turbine Scour

In addition, a comparison between the proposed equations in the current study for scour around Darrieus tidal current turbine and experimental data studied by Hill et al. [19] for horizontal axis turbine is presented in Figure 14. All the experiments had approached scour equilibrium after about 180 min. In Hill's experiment, the scour hole develops deeper and faster with upstream installed rotor. For turbine rotors installed upstream, the maximum scour depth was ~2.4$D$. The temporal scour depth indicated similar pattern with live bed scour type in our proposed equation. For downstream installed turbine rotor, the maximum scour depth was about 1.5$D$. This data was approximately same as $C/H = 1$ in our experiment. This was clear water type in our proposed equations. However, the temporal evolution of scour depth before scour equilibrium in downstream rotor condition in Hill's experiment was not match well with the current experiment. This may be caused by the different type of tidal current turbine.

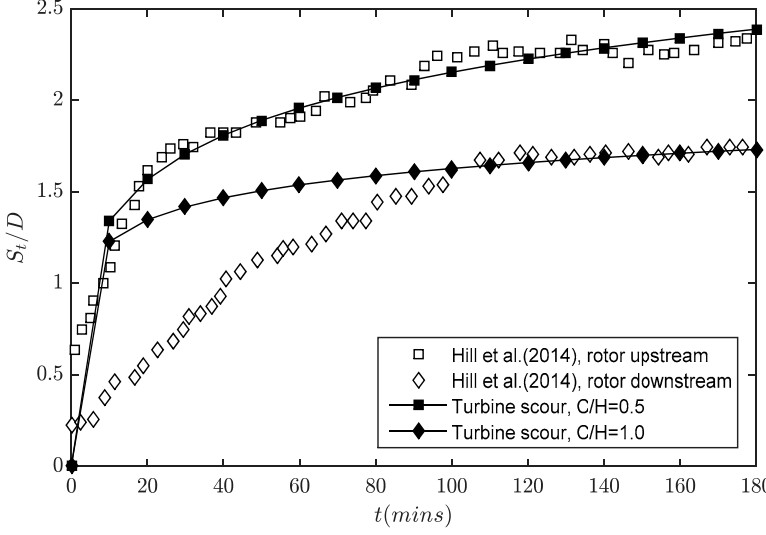

**Figure 14.** Comparison of time-dependent scour depth around two types of tidal current turbine.

## 5. Conclusions

The temporal evolution of seabed scour around monopile foundation of a Darrieus-type tidal current turbine is presented in the current study. The temporal scour depth and scour profiles are obtained by experimental measurement using 3D printed models with different rotor radius and tip clearance. Based on experimental data and analysis, empirical equations for temporal evolution of scour depth around turbine is proposed. By comparison analysis with other existing equations for other scour types, the influence of turbine rotor and tip clearance on the development of seabed scour

is discussed. The evidence of turbine rotor accelerates the mean velocity and increases the turbulence intensity between turbine and seabed. The scour hole size is amplified. The conclusions are as follows.

1.  The empirical equation for predicting the temporal evolution of scour hole depth around the turbine is proposed. In this equation, the turbine radius and tip-bed clearance are the two main influence factors for the turbine as these two factors can influence the hydrodynamic of turbine [7]. With the comparison with experimental data and existing equations, the empirical equation shows great ability to predict the temporal scour depth around tidal current turbine.

2.  Rotating turbine gives significant impacts to the turbine induced scour. (a) Tip-bed clearance: The equilibrium scour depth is deeper and evolution speed of scour hole is faster when tip clearance decrease in general. However, the size of scour hole is no longer increasing with smaller tip clearance when *C/H* is lower than 0.5. This means that live bed scour occurs when tip clearance is lower than 0.5*H*. In such conditions, tip clearance achieves the greatest impact on scour. (b) Rotor radius: the size of scour hole shows nonlinear relationship against rotor radius. The equilibrium scour depth decreases first and then increases when turbine rotor increases from 37.4 mm to 56.3 mm. The temporal evolution speed shows the same trend; (c) Compared with other types of scour: the scour hole around turbine foundation is bigger than scour at piers but smaller than scour induced by propeller. Our proposed equations can fit the temporal evolution of scour hole well.

3.  The temporal seabed profiles in the streamtube direction around the turbine foundation are plotted. The locations and values of equilibrium scour hole depth at different turbine radius and installation height are investigated. The maximum scour hole depth takes place at location about $x/D = -0.5$, $y/D = 0.5$. Generally, the scour hole size shows an inverse correlation with the space between turbine and seabed. The scour process can be divided as live bed scour or clear water scour due to different tip clearance. The amplification of scour depth is ~10–80% in different cases compared with local scour at piers. Furthermore, the scour hole shows rapid development in the first 10 min of the scour process, and then the scour depth grows slowly until equilibrium. These findings are valid for different tip clearance.

Our empirical model for temporal scour induced by Darrieus-type tidal current turbine has some limits. The model mainly considers the impact of tip clearance and rotor radius. Other influential parameters for scour, like sediment diameter and inlet velocity are considered as conventional conditions in our model. In these general conditions, our model shows great agreement with existed experimental data. However, there is some errors using the equation to calculate temporal scour development in some extreme conditions. In the future, it is suggested to study other parameters affecting the temporal and spatial development of scour induced by tidal current turbine.

**Author Contributions:** W.H.L., long term research series in tidal turbine and ship propeller jet induced scour with former supervisors M.D. and G.H.; C.S. and W.H.L. contributed to establish temporal scour equation of Darrieus type tidal turbine; C.S. wrote the manuscript with recommendations and validations from W.H.L., S.S.L., M.D. and G.H.

**Funding:** This research was funded by Natural Science Foundation of Tianjin City: 18JCYBJC21900 and The APC was funded by Natural Science Foundation of Tianjin City: 18JCYBJC21900.

**Acknowledgments:** The authors wish to extend their gratitude to School of Civil Engineering in Tianjin University for laboratory space; Queen's University Belfast, University of Plymouth, University of Oxford, Dalian University of Technology, University of Malaya (HIR ENG47), Universiti Teknologi Malaysia, and Southern University College for their past support; and professional bodies EI, IEI, IET, BCS, IEM, IEAust, ASCE, ECUK, BEM, FEANI, AFEO, MINDS and academy AAET for membership support and available resources.

**Conflicts of Interest:** The authors declare no conflict of interest.

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
