# Peer review of "Temporal Evolution of Seabed Scour Induced by Darrieus-Type Tidal Current Turbine"

_water, doi:10.3390/w11050896_

Round 1

Reviewer 1 Report

The paper shows a very interesting experimental approach to sea bed scour. The main finding is that the scour speed increases with the existance of a rotating turbine on the top of a monopile foundation. This seems to be obvious due to increased turbulences around the turbine in the flow velocity field. The investigations proves this derivation qualitatively.

The introduction gives a good overview on the state of the art. However, the description of the methodoly needs improvement, since this is necessary to understand the results. The explanation of the test setup is too short. In line 120 page 3 the reference [22] ist given for further details but the paper should be understandable by itself. Necessary additional information is needed for figure 1. It would be significant better to show all relevant cross sections and dimensions of the test setup instead of the picture of fig. 2. Very important for the paper is the introduction of the coordinates in figure 1. Furthermore v seems to be equal to Uc. One parameter should be used through the whole paper.

The uniformity of the flow was uggested but not controlled or shown. The symmetry of the scour could give a hint how good the uniformity is. However the 3D shape of the scour is not given. This would be an added value to the results. When an how was the laser used? Do sediments in the water affect the measurements?

Another important missing point is the discussion of the physical similarity relationships. The geometric scale for the turbine is given to 1:60. For the grain size d50=1.1 mm this leads to d50=66mm in nature. Which is gravel. The grain size distribution must be given, since its shape influences the erosion effects. A least the uniformity index shall be given. Furthermore boundary effect of the walls to the flow distribution should be discussed. Similarity theory should be applied to the used flow velocities and the realtionship to nature conditions.

The results should show y/D scour shapes also (not only two points). How much % of the bed width was eroded? Was there an influence from the rigid bottom or not?

p7 line 206. Figure 1 must be figure 5.

General statements as given on page 9 line 252 should be limited to the investigated boundary conditions. Still these are model tests.

Time dependent results will be dependent on the flow velocity (Figure 8). This should be critically discussed with perspective to the theories of similarity. The rotor speed will also influence the results. Please discuss that.

Figure 10 shows that many changes will happen in the first 10 minutes. A zoom into the first 10 minutes would be very interesting to understand the effects.

Figure 12 only shows two comparisons. This is too few to proof the accuracy of the model. Please perform more tests (outlook?).

page 15 conclusions.

The findings will depend a lot on the boundaries from the test set up. Caution is necessary when generalizing and transfering the findings.

General on format: Please review spacings. line 33, 49, 79, 87, 152 (brackets?) and line 396

Author Response

Responses to Reviewer#1’s comments:

Comment 1: The introduction gives a good overview on the state of the art. However, the description of the methodology needs improvement, since this is necessary to understand the results. The explanation of the test setup is too short. In line 120 page 3 the reference [22] ist given for further details but the paper should be understandable by itself. Necessary additional information is needed for figure 1. It would be significant better to show all relevant cross sections and dimensions of the test setup instead of the picture of fig. 2. Very important for the paper is the introduction of the coordinates in figure 1. Furthermore v seems to be equal to Uc. One parameter should be used through the whole paper.

Response 1: The description of the experimental methodology has been improved with more details of experimental set up and measurement methods in section 2. All relevant cross sections and dimensions of the test setup can be found in Figs.1-2. In these figures, the Uc is the unified representation of inlet velocity.

Comment 2: The uniformity of the flow was suggested but not controlled or shown. The symmetry of the scour could give a hint how good the uniformity is. However the 3D shape of the scour is not given. This would be an added value to the results. When an how was the laser used? Do sediments in the water affect the measurements?

Response 2: The uniformity of the flow was controlled by the recirculating flume and flow-equalizing equipment. The specific method and discussion has been added in section 2.1 and section 2.2. In addition, the 3D shape of the scour is given in Fig.3 to show how good the uniformity is. A sand dune is formed right behind the supporting pile of turbine and the dune shows great symmetry. This results can certify that the flow velocity has satisfactory uniformity. In addition, lines 123-133 are added to show how and when the laser used. In our experiment, the current velocity is not fact. It’s clear water scour in main scour area. So there is little sediment moving in water. Hence the sediments didn’t affect the measurements during the experiments.

Comment 3: Another important missing point is the discussion of the physical similarity relationships. The geometric scale for the turbine is given to 1:60. For the grain size d50=1.1 mm this leads to d50=66mm in nature. Which is gravel. The grain size distribution must be given, since its shape influences the erosion effects. A least the uniformity index shall be given. Furthermore boundary effect of the walls to the flow distribution should be discussed. Similarity theory should be applied to the used flow velocities and the relationship to nature conditions.

Response 3: The discussion of the physical similarity relationships is added in section 2.2. the geometric scale of turbine is approximately 1:60 of full scale Darrieus type tidal current turbine project like Kobold. The scaling effects of experimental set up is discussed in section 2.2.2. The reason how and why we choose grain size d50=1.1mm is illustrated in lines 164-170. Since the grain size is not the main variable to study the turbine’s impact of scour, we choose 1.1mm sediment size as a normal prevailing conditions.

Furthermore boundary effect of the walls to the flow distribution is discussed in lines 179-195 in section 2.2.2. The wall boundary shows little impact on sediment scour based on previous study, which has been stated detail. In addition, the physical similarity theory applying to the used flow velocities and relationship to nature conditions is added in section 2.2.1. In our experimental set up, the size, turbine solidity and tip speed ratio have been ensure to be in line with reality. And the Reynolds number is big enough to research the impact of turbine rotor. So our study can investigate the impact of turbine on its induced scour.

Comment 4: The results should show y/D scour shapes also (not only two points). How much % of the bed width was eroded? Was there an influence from the rigid bottom or not?

Response 4: The temporal development of scour width is shown in Fig.10. Final scour hole width is located between 3.5D to 4.8D with different rotor radius and tip clearance. The width data is about 2 times than scour depth in each case, and this is due to sand slide discussed in section 3.3. It is not an influence from the rigid bottom. It’s also the result of sediment scour.

Comment 5: p7 line 206. Figure 1 must be figure 5.

Response 5: It has been revised.

Comment 6: General statements as given on page 9 line 252 should be limited to the investigated boundary conditions. Still these are model tests.

Response 6: We added “According to model tests” before the sentence. However, the maximum scour depth position agrees with previous researches as listed in section 3.2.3. Since the maximum bed shear stress occurs at back sides of the monopile, the deepest position of scour hole appears here.

Comment 7: Time dependent results will be dependent on the flow velocity (Figure 8). This should be critically discussed with perspective to the theories of similarity. The rotor speed will also influence the results. Please discuss that.

Response 7: The discussion of the impacts of flow velocity, rotor speed have been added in section 3.3. Based on previous research paper, the faster inlet velocity can increase the scour depth. But the scour depth doesn’t increase more when the inlet velocity is more than critical velocity. Our main purpose is to find the impact of turbine on its induced temporal development of scour depth, so the constant inlet velocity is set up. However, the flow velocity below turbine rotor can be accelerated by turbine. With different hydrodynamic performance due to different tip clearance and rotor radius, the scour depth can be under great influence. Furthermore, the rotor speed can impact the results indeed, but we don’t focus on its impact in this paper. Hence we ensure the constant rotor speed in each test. The similarity theory of rotor speed is same tip speed ratio, which has been discussed in section 2.2.1.

Comment 8: Figure 10 shows that many changes will happen in the first 10 minutes. A zoom into the first 10 minutes would be very interesting to understand the effects.

Response 8: The description and discussion of temporal scour development especially in the first 10 minutes are added in section 3.3. The scour hole develops rapidly in the beginning process and then grows gradually until 150mins. This is a general appearance of scour around piles. Turbine scour starts with an increase of flow speed around turbine supporting pile. The sediment is easily washed away by large bed shear stress on both sides of pile. The scour hole expands by the energy of downflow and horseshoe vortex. However, the less bed shear stress and weaker energy of horseshoe vortex exists inside the hole along with the development of scour hole until the equilibrium situation.

Comment 9: Figure 12 only shows two comparisons. This is too few to proof the accuracy of the model. Please perform more tests (outlook?).

Response 9: There are a few of researches about tidal current turbine scour. There is even no other paper focus on scour induced by tidal current turbine. So we can only find two comparisons of turbine scour depth to prove the accuracy of our model. However, we choose many data about two other types of scour: scour around pile and scour induced by propeller to validate our model. The results show that the existence of turbine can increase the scour depth compared to scour around single pile. These comparisons can also prove the rationality of our proposed model.

Comment 10: Page 15 conclusions. The findings will depend a lot on the boundaries from the test set up. Caution is necessary when generalizing and transfering the findings.

Response 10: We added a paragraph in the end of section 5, Conclusion. The cautions about our findings and the proposed equation are illustrated in that paragraph. At last, the suggested future work is presented.

Comment 11: General on format: Please review spacings. line 33, 49, 79, 87, 152 (brackets?) and line 396

Response 11: Thanks for the comment about detail format, we have revised them.

Reviewer 2 Report

Please revise the manuscript according to the attached file.

Author Response

Responses to Reviewer#2’s comments:

Comment 1: Lines 35 and 36: The Horseshoe vortex has been introduced as the only dominant factor of scour around the piles with tidal current turbine, similar to a single pile without any rotor. Do the authors think that the influence of a rotor is really significant? To what extent?

Response 1: we added a paragraph in lines 36-40 to illustrate the influence of horseshoe vortex on turbine scour. The influence of rotor is also discussed in these sentences based on previous studies. Actually, the influence of a rotor is really significant for scour induced by turbine, since the existence of rotor accelerate the bed velocity and then amplify the temporal and spatial development of scour depth.

Comment 2: Lines 50 to 56 the role of horseshoe vortex has been highlighted; however, the importance of wake vortices was not discussed. Please emphasize the role of wake vortices on scour development.

Response 2: During the scour development, the wake-vortex system can expand the scour hole downstream. The wake-vortex system acts like a shovel to remove the bed material which is then carried downstream by shedding eddies from the supporting piles. We have added lines 57-61 to emphasize the role of wake vortices on scour development.

Comment 3: Lines 57 to 60: "The scour process..." is repetitive. You have discussed it in the previous sections. Please remove it from the manuscript.

Response 3: We have removed it from the manuscript.

Comment 4: Line 63: Please add more details about the findings of Chen and Lam. For example, did they simulate the scour evolution or only bed shear stress variation?

Response 4: We have added more details about the findings of Chen and Lam in lines 65-67. They found the presence of turbine rotor changed the boundary layer profile and results in the altering of horseshoe vortex formation. The axial velocity of flow increased about 10% of initial velocity which can cause deeper scour hole.

Comment 5: Line 71: Please add more explanation about the "vortex method" in the manuscript. Is it a numerical method?

Response 5: We have added explanation about the “vortex method” in lines 74-77. The “vortex method” was a velocity-vorticity numerical implementation of the Navier-Stokes equations to compute an unsteady evolution of the turbine wake by some three-dimensional software.

Comment 6: Line 73: Please replace “an wake equation” with “a wake equation”.

Response 6: We have replaced these words in line 78.

Comment 7: Line 102: the channel width is narrow (=35 cm). In this condition, secondary currents may develop and affect the experimental results. Have you considered this problem?

Response 7: Actually, the channel width (875px) is not such narrow in our experiments. And no secondary currents developed in the experiments. Furthermore, the flow-equalizing equipment works well to ensure the uniformity of flow. We added a section to discuss the scaling effects of experimental set up in section 2.2.2, lines 164-204  including the discussion of channel width. In fact, the channel width is about 6 times than turbine radius, 35 times than supporting pile diameter. This size ratio is almost same as Roulund’s experiment of scour around piles. And previous’ work of flow field around tidal current turbine showed that boundary acted little impacts on turbine’s surrounding flow field. In addition, Fig.3 of final sediment bed profiles showed great symmetry to prove the good uniformity of flow.

Comment 8: I suggest adding the following sentence at the end of line 107 after “or y-axis direction.”:

“Laser-based instruments provide accurate results when they are utilized to acquire scour pattern. For instance, Dodaro et al. (2014) and Dodaro et al. (2016) employed a 3D laser scanner to measure scour profile downstream of a rigid bed."

Then, please add the following articles into the ‘References’:

- Dodaro, G., Tafarojnoruz, A., Stefanucci, F., Adduce, C., Calomino, F., Gaudio, R. and Sciortino, G. (2014). “An experimental and numerical study on the spatial and temporal evolution of a scour hole downstream of a rigid bed.” Proc. the International Conference on Fluvial Hydraulics, 1415-1422.

- Dodaro, G., Tafarojnoruz, A., Sciortino, G., Adduce, C., Calomino, F., Gaudio, R. (2016). “Modified Einstein sediment transport method to simulate the local Scour evolution downstream of a rigid bed.” Journal of Hydraulic Engineering, 142(11), DOI: 10.1061/(ASCE)HY.1943-7900.0001179.

Response 8: Thanks for the advice, we think these two paper prove the rationality of using Lase-based instruments to acquire scour pattern. So we have added these two paper as reference.

Comment 9: Line 130: It is stated that a scale of 1:60 was respected in construction of the model. Have you considered a real Darrieus type current turbine?

Response 9: Yes, we considered a real Darrieus type current turbine named Kobold. In our experiment, the turbine size is approximately 1:60 of this turbine constrained by the dimensions of flume. We added a paragraph in lines 144-154 to describe the physical similarity relationships of our model.

Comment 10: Lines 142 to 144: Please add a reference for the statements.

Response 10: We have added the reference for that statements.

Comment 11: Lines 149 and 150: In general, it is unlikely to obtain equilibrium condition after 150 minutes. Longer duration is needed under clear-water scour condition to achieve equilibrium. Can you present a figure that shows temporal variation of maximum scour depth for a relatively long test (much longer than 150 minutes, e.g., one day)?

Response 11: In our experiments, the scour depth and scour width show no more increase after about 150 mins later. This can be seen in Figs.9-10. In fact, we did the test for a relatively long time about 6 hours, and the final scour depth is approximately similar as other tests presented in the paper. Furthermore, based on previous researches, the scour can reach equilibrium when the flow can no longer wash away the sediments inside scour hole. And their experiments also shown the equilibrium time for clear water scour is about 1-2 hours later. The results can be found in Fig.12 and Fig.14. To be more accurate, we changed this sentence to “Seabed scour reached equilibrium after about 150mins in the experiments”.

Comment 12: Lines 173 and 174: Do the authors think that the scour depth reach 75% of final depth after 30 minutes regardless of the mean approach flow velocity? I think that equilibrium scour depth may obtain in a shorter duration for a larger flow velocity.

Response 12: In our model tests, the temporal scour depth can reach more than 75% of final depth after about 30 mins. The increase rate of scour depth slows down with time. In our experiments and other previous works, the scour depth increase fast in the initial stage. By the way, we didn’t say the scour depth reach 75% of final depth after 30 mins regardless of flow velocity. The equilibrium scour depth may develop faster for a larger flow velocity, but it’s not the main research point in the paper. So we used same flow velocity in all tests to study the impacts of turbine on the temporal development of scour. Furthermore, it is an interesting idea to investigate the duration time to equilibrium scour depth for different flow velocity, we may research it in the future.  

We changed to sentence to “Temporal scour depth can reach 75% of final depth after initial 30mins in the model tests.”

Comment 13: Lines 286 to 288: Regarding the size and depth of pier scour with respect to the rotating turbine scour, can the author provide a rough number indicating that to what extent a scour caused by a rotating turbine is larger than a pier-induced scour?

Response 13: The increase ratio is about 10%-80% with various turbine rotor radius and tip clearance in the model tests. As shown in line 427, the equilibrium scour depth around single pile in same test conditions is 1.45D, and the scour depth with turbine rotor is from 1.6D to 2.6D. The reasons why the scour depth induced by rotating turbine is deeper than pier-induced scour were illustrated in section 3.3 in lines 335-345.

Comment 14: I suggest adding the following sentence at the end of line 336 after “…checking the accuracy.”:

"According to Gaudio et al. (2010) and Gaudio et al. (2013), due to different mathematical structures in scour prediction formulas, available bridge pier scour equations may give significantly different predictions."

Then, please add the following papers into the ‘References’:

- Gaudio, R., Grimaldi, C., Tafarojnoruz, A., Calomino, F. (2010). “Comparison of formulae for the prediction of scour depth at piers.” Proc. 1st European IAHR congress, Edinburgh, UK, 6 pp.

- Gaudio, R., Tafarojnoruz, A., De Bartolo, S. (2013). "Sensitivity analysis of bridge pier scour depth predictive formulae." Journal of Hydroinformatics, 15(3), 939-951, DOI: 10.2166/hydro.2013.036.

Response 14: Thanks for the suggestion. We had added these to papers as references to strengthen the scientific nature of comparison and discussion in section 4.2.1.

Comment 15: Lines 337: Why have you set Se =1.45D?

Response 15: As we presented in line 427, the Se=1.45D is the equilibrium scour depth around single pile for the same test conditions measured in the current experiments. So we set Se = 1.45D to compare the temporal development of scour depth between turbine induced scour and pier induced scour.

Comment 16: Lines 391 and 392: I think the statement "the turbine radius ... for turbine." is not your own finding. If it is a finding of any previous study, please cite that study.

Response 16: This is actually our own finding because there is no more paper talks about the scour induced by Darrieus type tidal current turbine. Our experiments confirmed that tip clearance and rotor radius can influence the scour development greatly. Anyway, we added a sentence to illustrate why we choose these two parameters in our equation and cited the reference.

Comment 17: Line 393: I think only one “sediment size” was used in your study. If yes, you cannot consider "sediment size" as an influencing parameter in your conclusions.

Response 17: We deleted that sentence since we didn’t test the impact of different sediment size on scour depth.

Comment 18: Line 402: please replace “fist” with “first”.

Response 18: Thanks for the careful read, we have replaced that.

Comment 19: Line 402: please replace “increase” with “increases”.

Response 19: we have replaced that.

Comment 19: Line 402: please replace “increase” with “increases”.

Response 19: we have replaced that.

Comment 20: Line 412 and 413: Is the indicated finding valid for different tip clearance? Please clarify within the paper.

Response 20: The indicated finding is valid for different tip clearance. This is a main factor we considered in the experiments and empirical equation. We have clarified it within the paper in line 503. 

Round 2

Reviewer 2 Report

The paper can be published in present form.